# Synthesis of Novel Shape Memory Thermoplastic Polyurethanes (SMTPUs) from Bio-Based Materials for Application in 3D/4D Printing Filaments

**DOI:** 10.3390/ma16031072

**Published:** 2023-01-26

**Authors:** Yang-Sook Jung, Sunhee Lee, Jaehyeung Park, Eun-Joo Shin

**Affiliations:** 1Department of Organic Materials and Polymer Engineering, Dong-A University, Busan 49315, Republic of Korea; 2Department of Biofibers and Biomaterials Science, Kyungpook National University, Daegu 41566, Republic of Korea; 3Department of Fashion Design, Dong-A University, Busan 49315, Republic of Korea

**Keywords:** shape memory polymer, thermoplastic polyurethane, bio-based material, shape recoverability

## Abstract

Bio-based thermoplastic polyurethanes have attracted increasing attention as advanced shape memory materials. Using the prepolymer method, novel fast-responding shape memory thermoplastic polyurethanes (SMTPUs) were prepared from 100% bio-based polyester polyol, poly-propylene succinate derived from corn oil, diphenyl methane diisocyanate, and bio-based 1,3-propanediol as a chain extender. The morphologies of the SMTPUs were investigated by Fourier transform infrared spectroscopy, atomic force microscopy, and X-ray diffraction, which revealed the interdomain spacing between the hard and soft phases, the degree of phase separation, and the intermixing level between the hard and soft phases. The thermal and mechanical properties of the SMTPUs were also investigated, wherein a high hard segment content imparted unique properties that rendered the SMTPUs suitable for shape memory applications at varying temperatures. More specifically, the SMTPUs exhibited a high level of elastic elongation and good mechanical strength. Following compositional optimization, a tensile strength of 24–27 MPa was achieved, in addition to an elongation at break of 358–552% and a hardness of 84–92 Shore A. Moreover, the bio-based SMTPU exhibited a shape recovery of 100%, thereby indicating its potential for use as an advanced temperature-dependent shape memory material with an excellent shape recoverability.

## 1. Introduction

Shape memory polymers (SMPs) belong to a well-known class of intelligent or smart materials and have attracted considerable scientific attention over the past decade owing to their technological significance for a wide range of applications, including in biomedical devices, microelectromechanical systems, aerospace apparatuses, actuators, complex four-dimensional (4D) printed structures and devices, and smart textiles [1,2]. SMPs are known to adopt temporary shapes upon the application of an appropriate stimulus, such as a temperature change, the irradiation of light, exposure to moisture, a change in pH, or the application of a magnetic or electric field [3]. Moreover, the unique shape-changing properties of SMPs lead to technical advantages owing to their good recovery behaviors, and their relatively large and fast response performances within a narrow range of temperatures [4]; these properties surpass those of well-known metallic shape memory alloys. In addition, SMPs exhibit characteristics of both hard and soft materials, and their elasticity modulus shows a reversible change in relation to the transition temperature. In terms of the shape memory effects of SMPs, these are based on the existence of two structural elements, namely permanent net-points and reversible molecular switching segments [5]. The permanent net-points create a three-dimensional (3D) network architecture that can be based on different chemical structures, crosslinks, crystal segments, hydrogen bonding, and macromolecular entanglements [4,6], and this network can impart memory/storage properties to the system. In terms of the reversible molecular switching segments, these moieties help to fix or release the temporary shapes below the transition temperature, wherein either the glass transition temperature or the melting temperature serves as the transition/switching temperature [7]. As a result, temporary shapes can be formed above the transition temperature and can be fixed by cooling down below the transition temperature; the permanent shape can be obtained once again by heating above the transition temperature. The shape memory transformation, therefore, depends on the transposition of polymer molecules between the constrained and random entangled conformations. Consequently, in terms of the chemical structure, SMPs can be considered as phase-separated linear block copolymers with hard and soft segments, wherein the hard segment contains the permanent net-points, and the soft segment is responsible for the reversible phase transformation [6,7]. As a result, this molecular mechanism leads to shape memory properties based on programming of the temporary form and recovery of the permanent shape.

Many polymers have been developed that are capable of exhibiting shape memory effects, such as those based on polyethers, polyacrylates, polyamides, polysiloxanes, polyurethanes, polyether amides, polyvinyl ethers, polyurethane/ureas, polyether esters, and urethane/butadiene copolymers [8,9]. To date, the development of temperature-dependent SMPs has focused primarily on thermoplastic polyurethanes (TPUs), crosslinked polyethylene, poly(lactic acid) (PLA), poly(glycolic acid) (PGA), poly(ε-caprolactone) (PCL), sulfonated ethylene–propylene–diene terpolymer (EPDM), and polynorbornene, along with their co-polymers or blends [10,11,12,13]. Among the known SMPs, segmented TPUs are of particular interest, and a large number of research articles and reviews have been published in this area during the past few decades [14]. Segmented TPUs are prepared from three basic starting materials, including a long chain polyol (the soft segment), diisocyanate, and chain extender, wherein the latter two components form the hard segment. These types of polyurethanes are characterized by segmented structures (i.e., block copolymers), and their morphologies depend on their chemical compositions along with the chain length of the soft segment; these factors are also responsible for determining their shape memory properties [4,5]. In addition, the quantity of hard segment affects the recovery ratio, wherein a low content leads to incomplete recovery of the deformed specimen [1,5]. The recovery is also affected by the modulus ratio and the size of the dispersed phase [8]. In this context, Xu et al. developed a system based on polypropylene glycol (PPG) with 4,4′-diphenylmethane diisocyanate (MDI) and 1,4-butanediol (BDO) as the hard segment. Despite the excellent mechanical and shape memory properties of their product, the transition temperatures for shape-changing ranged from −12.5 to −5.2 °C, thereby rendering this system unsuitable for stable secondary shape fixation in biological devices (i.e., at physiological temperature) [15]. In addition, Hasirci et al. synthesized segregated TPUs that exhibited shape memory properties [16], while Xu et al. [17] and Peng et al. [18] incorporated PPG into a system based on using MDI and BDO as the hard segment. The resulting products exhibited high transition temperatures and appropriate mechanical and shape memory properties. Furthermore, Candau et al. produced a strain-induced shape memory thermoplastic polyurethane (SMTPU) based on poly(ethylene oxide) (PEO; soft segment), bis(4-isocyanatocyclohexyl) methane (H_12_MDI; diisocyanate), and 2-methyl-1,5-diaminopentane (MDAP; chain extender) as the hard segment [19]. They also found that using PPG as the soft segment and triethylene glycol as the chain extender with hexamethylene diisocyanate led to polymers exhibiting a shape recovery ability of >90% in addition to sufficiently high transition temperatures (56–58 °C). Moreover, Lai et al. reported shape fixing ratios of >96.3% and shape recovery ratios of >91.2% for TPU/olefin block copolymer (OBC)/PCL blends [20], while Tamara et al. used a diisocyanate derived from L-lysine and a macrodiol derived from castor oil to prepare thermally-responsive, crosslinked, and bio-based polyurethanes [21]. In another study, Zhu [22] designed an SMTPU based on poly(ε-caprolactone diol), MDI, and BDO for use as a soft artificial muscle actuator capable of realizing two main contractions. It was shown that the degree of contraction was easily tuned by altering the crystallinity of the soft block. They reported that the sensor abilities of the synthesized polymers depended on the temperature and, thus, on the quantity of crystalline phase that was present.

To date, SMTPUs have been used in various applications, including adhesives, textile and apparel products, medical devices, and soft actuators, owing to their versatile structures and properties, such as their relatively low cost, simple synthetic methods, good processability properties, and high stability to the surrounding atmosphere, in addition to their convenient shape-memory temperature intervals and high degrees of shape recovery [23]. More specifically, polymer temperature sensors have been developed, as have SMTPUs, for use in the handgrips of spoons, toothbrushes, and kitchen knives. SMTPUs are also widely employed in the medical field owing to their engineered ability to soften at body temperature. As a result, they have been applied in self-retaining needles, such as those required for drips and catheters [24,25]. Degradable polymer implant materials that exhibit shape memory effects have also been developed for applications in orthodontic materials, bone screws, prosthetics, catheters, tubes, films, stents, orthopedic braces, splints, tapes for preparing casts, and implants [1,3]. It is also possible to apply SMTPUs to textiles, apparels, and related products, such as insulating fibers, breathable fibers with shape memory properties, shoes, and wrinkle/shrink-resistant finishes for apparel fabrics. Thus, due to their favorable properties, it is expected that the application of SMTPUs could be easily extended to other areas.

TPUs can be easily processed for use in casting, injection-molding, and extruded filaments for 3D/4D printing [26], which is of particular interest due to the rapid development of the 3D/4D printing industry. Among the 3D/4D printing materials reported to date, the TPU filaments have been reported to exhibit an excellent elasticity and flexibility, in addition to good shock absorption properties. TPU materials have been used previously in energy absorption, soft-robotics, drug delivering devices, biomedical devices, tissue engineering, mechanical actuators, and prosthetics application [22,27,28,29,30,31,32]. For instance, Bates et al. [27] developed a parametric study that captured the energy absorbing capability of hexagonal arrays manufactured from two types of TPU, with relative densities of 0.18–0.49. Yap et al. [30] presented a novel technique for the direct 3D printing of soft pneumatic actuators using 3D printers based on fused deposition modeling (FDM) technology with thermoplastic elastomer filament (Shore hardness of 85 A). More recently, Rahmatabadi’s group [32] has proposed the combination of the microscopic concept of SMPs and multi-material printing of a thermoplastic elastomer using FDM to create a new 4D printing method without SMP and the additional operations such as synthesis and blending. In another study, Kızıltay et al. [31] synthesized thermoplastic poly(ester-urethane) (PEU) from L-lysine diisocyanate (LDI) and polycaprolactone diol (PCL) for tissue engineering applications. However, due to environmental concerns, the demand for naturally-derived filament materials has increased. Due to the impurities present in natural raw materials, material processing becomes more challenging, yellowing can occur, and air bubbles can be generated, ultimately leading to a high defect rate, and reduced physical properties (e.g., strength, elongation, and hardness) compared to synthetic materials [33].

Thus, we herein report the preparation, properties, and application of bio-based SMTPUs. Since the use of bio-based polyols could lead to novel properties, we also investigate ester-based polyurethanes and the effect of the [NCO]/[OH] ratio on their structures, microphase separation, morphologies, rheological properties, thermal properties, and mechanical properties. In addition, the development of environmentally friendly, temperature-dependent SMPs for 4D printing is examined in terms of the combined properties of the polymer components and morphologies, wherein a number of key factors are considered (e.g., glass transition temperature, melting temperature, ductility, melting point, viscosity, crystallinity, hardness, and tensile properties).

## 2. Materials and Methods

### 2.1. Materials

The hard segment was prepared using MDI and bio-based 1,3-propanediol (PDO, chain extender). These two components were purchased from Sigma-Aldrich (Burlington, MA, USA) and Dupont Tate & Lyle BioProducts (SUSTERRA (R) PROPANEDIOL, Wilmington, DE, USA), respectively. The bio-based polyol, polypropylene succinate (PPS) was kindly donated by Dong Ah Chemical (Busan, Republic of Korea). The PPS was produced from corn oil (100% bio-based content) and had an average molar mass (Mn) of 1000 g/mol. T9 (tin(II) 2-ethylhexanoate) was purchased from Sigma-Aldrich (Burlington, MA, USA) and was used as a catalyst. All materials were used as received without further purification.

### 2.2. Synthesis of the Bio-Based Shape Memory Thermoplastic Polyurethanes

Bio-based SMTPU with different hard section contents were prepared via the pre-polymer polymerization process with variation in the [OH]/[NCO]/[OH] ratio (see Figure 1), wherein the first [OH] refers to the polyol OH content, [NCO] refers to the NCO groups of MDI, and the second [OH] corresponds to the PDO OH content. Thus, maintaining the [NCO]/[OH] molar ratio at 1:1, a selection of SMTPUs were prepared, as outlined in Table 1. More specifically, the MDI was melted in an oven at 80 °C for 4 h then added to the PPS. After stirring the resulting mixture at 250 rpm and 80 °C for 1 h in a nitrogen atmosphere, the PDO was added and stirring was continued for 1 to 5 min. Subsequently, the T9 catalyst was added, and the mixture was stirred at 200 rpm for a further 1 min. After this time, the desired SMTPU was obtained by drying at room temperature (20–25 °C) for 24 h, drying in a vacuum dryer at 80 °C for 20 h, and curing to complete the polymerization process. The hardened SMTPU was then pressed at 200 °C for 3 min using a hydraulic press (CARVER, Wabash, IN, USA) to obtain a film. After cooling to room temperature, the 0.5–1 mm thick sheet was removed from the mold and used for chemical structural, thermal, and mechanical analysis, in addition to shape memory characterization.

### 2.3. Preparation of the Bio-Based Shape Memory Thermoplastic Polyurethanes Filaments

A single-screw extruder (FILIBOT^®®^ H400, Fordentech Co., Seoul, Republic of Korea) was used to obtain SMTPU filaments for 3D printing. The nozzle diameter was controlled at 1.5–1.7 mm. Combinations of different parameters were tested to obtain an SMTPU filament with stable diameter and dimensions, as outlined in Table 2. It should be noted that, upon increasing the content of the hard segment, it was possible to carry out extrusion at higher screw temperatures. During fabrication, 1.5 mm filaments were obtained with a screw speed of 13.0–13.2 rpm.

### 2.4. D Fused Deposition Modeling (FDM) Printing of Shape Memory Thermoplastic Polyurethanes Filaments

After extrusion of the TPU filament, an auxetic re-entrant (RE) was printed using the Cubicreator program (Cubicon Inc., Seongnam, Republic of Korea) and a fused deposition modeling (FDM) 3D printer (Cubicon Single Plus, TPC Mechatronics Corp., Incheon, Republic of Korea). In our previous study [9,34], a 3D-printed sinusoidal and auxetic sinusoidal pattern was prepared. In the current study, a RE sample measuring 10 mm × 10 mm, with a repeat size of 30 mm × 30 mm, has been developed. Subsequently, a 3D RE pattern with a thickness of 2 mm was obtained using Autodesk 123D for the .stl file and with the application of Cubicreator program software (Cubicon Inc., Korea). The nozzle temperature was controlled during the preliminary test before 3D printing at 190–230 °C. Finally, the SMTPU samples were 3D-printed at 230 °C using a print speed of 12 mm/s, a layer height of 0.2 mm, and an infill density of 100% in all cases.

### 2.5. Characterization

#### 2.5.1. Chemical Structures, Morphologies, and Crystalline Analyses of the Bio-Based Shape Memory Thermoplastic Polyurethane Films

Fourier transform infrared (FTIR) spectroscopy measurements were performed over the scan range of 400–4000 cm^−1^ using a Nicolet Nexus FTIR spectrometer (PerkinElmer, Shelton, CT, USA) equipped with an attenuated total reflectance (ATR) accessory. Atomic force microscopy (AFM, Inova system, Bruker, Billerica, MA, USA) was used to investigate the surface morphologies of the TPU films. For this purpose, the AFM instrument was equipped with a standard silicon nitride probe (Super Sharp Silicon™, SPM-Sensor, Na-noSensors™, Zurich, Switzerland), and the spring constant and resonance frequency were set at 42 N and 320 kHz, respectively. All measurements were performed under standard conditions using the AFM tapping mode technique. Surface images were recorded with dimensions of 20 × 20 µm, and the bulk morphology was measured by imaging the fracture area after previous freeze fracture of the sheets at −80 °C. The AFM images were processed using NanoScope analysis software. Other scanning parameters such as integral and proportional gains were automatically set by the Nanoscope software from Bruker Corporation (Bruker Application Notes, 2009). X-ray diffraction (XRD) measurements were carried out using a Shimadzu diffractometer (XRD-6000) with monochromatic CuKα radiation (λ = 0.15418 nm) at 40 kV and 30 mA. These measurements were carried out over a 2θ scan range of 5–40° with a step size of 0.05° and a rate of 2 s/step (i.e., 1.5° min^−1^). Peak separations were performed using Gaussian deconvolution.

#### 2.5.2. Mechanical and Thermal Analyses of the Bio-Based Shape Memory Thermoplastic Polyurethane Films

The dynamic mechanical properties of the prepared SMTPU films were evaluated using a dynamic mechanical analysis (DMA) Q800 analyzer (TA instruments, Newcastle, DE, USA) in the tensile mode at a frequency of 1 Hz. The samples were initially cooled to −100 °C, and then heated to 150 °C at a rate of 4 °C/min. The thermal properties of the prepared SMTPU films (2–10 mg) were evaluated using a differential scanning calorimetry (DSC) 8500 analyzer (TA Instruments, USA). These measurements were carried out from −70 to 250 °C with a heating rate of 20 °C/min under a nitrogen purge. Thermogravimetric analysis (TGA, Q500 TGA, TA Instruments, USA) was performed between 40 and 650 °C with a heating rate of 10 °C/min under a nitrogen atmosphere. The mass of each initial sample was ~5 mg. The weight losses at 5 and 50% were recorded, in addition to the maximum degradation rate and the ash residue at 600 °C. The Shore A hardness values were deter-mined at room temperature (Zwick Roell GS-706 N analogical hardness testing apparatus, TECLOCK, Nagano, Japan) according to standard UNE-EN ISO 868:1998: Plastics and ebonite. Determination of the indentation hardness was achieved using a Shore hardness durometer according to the standard procedure at (23 ± 2) °C and 50% relative humidity. The mechanical properties (i.e., the stress–strain behaviors) of the bio-based SMTPU films were evaluated on an Instron 4201 autograph tester (Shimadzu, Kyoto, Japan). For this purpose, specimens measuring 10 mm in length, 2 mm in width, and 0.5 mm in thickness were employed. Thermal–mechanical cyclic tensile tests were performed at 25 and 75 °C, and the crosshead speed was 20 mm/min. The limit of constant strain was 100% and the cyclic number was 10.

#### 2.5.3. Characterization of the Bio-Based Shape Memory Thermoplastic Polyurethane Filaments

The surface morphologies of the SMTPU filaments were observed using optical microscopy (NTZ-6000, NEXTVISION, Scottsdale, AZ, USA). The mechanical properties (i.e., the stress–strain behaviors) of the bio-based SMTPU filaments were tested on an Instron 4201 auto-graph tester (Simadzu, Japan). For this purpose, specimens measuring 10 mm in length and 1.4–1.5 mm in thickness were employed.

#### 2.5.4. Characterization of the Shape Memory Properties

The shape memory behaviors of the bio-based SMTPU films were evaluated using SMP program Ⅰ. More specifically, a glass rod was wrapped with the original film and placed in an oven at 70 °C (i.e., above T_trans_) for 5 min. The wrapped specimen was then allowed to cool at room temperature (i.e., below T_trans_) until the thermal stress disappeared and, subsequently, the wrapped specimen was heated at 75, 80, or 85 °C to recover its original shape. Subsequently, SMP program II was used to carry out the shape memory test, as outlined in Figure 2. In this case, the original length of the specimen (*ε_m_*) was recorded before extending at room temperature. The sample was then heated to 75 °C and held at this temperature for 5 min prior to extending the sample to a strain of 200% (*ε_μ_*), and cooling to room temperature. After releasing the tensile force, the unloaded strain (*ε_μ_*) was recorded. The stretched specimens were then heated at various recovery temperatures (50–90 °C) and held for 5 min to allow recovery to their original shapes, and the final residual strain (*ɛ_p_*) was recorded for each specimen. The tensile loading, unloading, and heating cycle was repeated three times using three specimens from each group. The shape recovery ratio (R_r_) was calculated according to the following equation [4]:$$R_{r}=\frac{\varepsilon_{m}-\varepsilon_{p}}{\varepsilon_{m}}$$

#### 2.5.5. Shape Memory Behaviors of the 3D-Printed SMTPUs

To investigate the shape memory behaviors of the 3D-printed SMTPUs, SMP program Ⅰ was employed. More specifically, a glass rod was wrapped with the original 3D-printed composite and placed in an oven at 75 °C for 5 min. The wrapped specimen was then allowed to stand at room temperature until the thermal stress disappeared. Subsequently, the wrapped specimen was heated to 80, 85, or 90 °C to recover its original shape. The time required to return to the original shape was measured.

## 3. Results and Discussion

### 3.1. Characterization of the Chemical Structures of the SMTPU Films

The chemical structures of the synthesized SMTPU films were initially examined using FTIR spectroscopy. As shown in the spectra presented in Figure 3, bands were observed for all materials corresponding to the characteristic groups of TPU [35]. More specifically, the N−H bond stretching vibration of the urethane groups appeared at 3320 cm^−1^, while the band at 1530 cm^−1^ was attributed to the N–H and C–N groups, and the band a 1257 cm^−1^ corresponded to the C–O stretching of the PPS ester groups [36]. As previously reported, the key structural feature of an SMTPU is its two-phase microstructure, wherein the degree of microphase separation is essential to achieving SMTPU elastomers with good mechanical properties [37]. In this context, the hydrogen bonding interactions are pivotal in determining the microphase separation and microstructure, which in turn influence the mechanical and thermal properties of the polymer [38]. In such systems, the incompatibility between segments leads to such microphase separation, and this can be controlled by variation in the block lengths and the degree of hydrogen bonding and crystallization; these factors also affect the shape recovery ability of the final material [39]. Thus, in terms of the FTIR results, the C=O stretching region (1680–1750 cm^−1^) can be used to quantitively identify the presence of hydrogen bonds [40]. However, in the current system, the C=O spectral region is particularly complex due to the presence of free C=O ester groups in the soft domain, free C=O groups in the hard domain, hydrogen-bonded C=O groups in the disordered hard domain, hydrogen-bonded C=O moieties in the ordered hard domain, and hydrogen-bonded C=O moieties in the disordered soft domain [41] that overlap with the peaks corresponding to the urethane C=O groups. Nonetheless, the band envelope was decomposed into five Gaussian bands by curve-fitting using Origin software (Origin 2018), and the contents of these different C=O regions in the prepared SMTPUs are given in Table 3. As indicated, the free C=O species is dominant in MP 1/1.50/0.50 (lower weight fraction of hydrogen-bonded urethane groups), and the hydrogen-bonded C=O groups of the ordered hard domain are dominant in MP 1/2.22/1.22 (higher weight fraction of hydrogen-bonded urethane groups).

The percentage of hydrogen-bonded C=O in the ordered hard domain at MP 1/2.22/1.22 was at a high level of 39%. In MP 1/1.50/0.50, the percentage of hydrogen-bonded C=O in the ordered hard domain was lower; however, the percentage of hydrogen-bonded C=O in the disordered soft domain was higher than the other samples.

Subsequently, the ratio of hydrogen-bonded C=O species on the urethane groups was employed to evaluate the degree of microphase separation, and is calculated as the ratio of hydrogen bonded to free urethane groups. Regarding the new deconvolution analysis, the weight fraction of the hydrogen-bonded urethane groups (*X_b_*) can be calculated according to Equation (1) [42]:


(1)
$$X_{b}=\frac{A\left(1720\;{\mathrm{cm}}^{-1}\right)+A\left(1700\;{\mathrm{cm}}^{-1}\right)}{k^{\prime }A\left(1730\;{\mathrm{cm}}^{-1}\right)+A\left(1720\;{\mathrm{cm}}^{-1}\right)+A\left(1700\;{\mathrm{cm}}^{-1}\right)\;}\;(k^{\prime }=1.2)$$


In addition, the weight fraction of the hard segment in the soft phase (*W′*_2_), mixed phase weight fraction (MP), soft phase weight fraction (SP), and hard phase weight fraction (HP) were also calculated for the prepared TPUs as per Equations (2)−(5) (*f*: the weight fraction of hard segment before synthetic process) [43]:


(2)
$${W^{\prime }}_{2}=\frac{\left(1-X_{b}\right)f}{\left[\left(1-X_{b}\right)f+\left(1-f\right)\right]\;},$$



MP = *f W*′_2_,
(3)



SP = MP + (1 − *f*),
(4)



HP = 1 − SP.
(5)


As can be seen from Table 4, all SMTPUs are characterized by high weight fractions of hydrogen-bonded urethane groups (*X_b_*), independent of the content of the hard segment. The data obtained for the synthesized bio-based SMTPUs using the new deconvolution method indicate that only ~0.05−0.99 wt% of the hard segment was present in the soft phase (*W′*_2_), thereby confirming that this segment was mainly located in the hard phase, and ultimately confirming the high degree of phase separation in all samples; this can lead to an increased number of hydrogen-bonded urethane groups. In addition, in the case of the MP 1/2.22/1.22 specimen, the *W′*_2_ was slightly reduced in correlation with the increasing content of the hard segment. In comparison to other TPUs described by Niemczyk et al. [44], Fuensanta [45], and Fernandez [42], which contained hard segment contents of 35–41 wt% and *W*′_2_ values of 0.11–0.14, 0.28–0.36, and 0.06–0.13, the MP 1/1.50/0.50 species prepared herein exhibited a higher *W*′_2_ value of 0.99.

Overall, these results indicate that the MP 1/2.22/1.22 specimen exhibited a good phase separation of the hard and soft segments, in addition to well-oriented domains in the hard segment. Furthermore, the MP 1/1.5/0.5 specimen clearly exhibited a relatively-low hard segment content, and, in this sample, a large amount of the hard segment was found in the soft segment, indicating a reduced degree of phase separation.

### 3.2. Morphology and Crystalline Properties

Since the [NCO]/[OH] molar ratio is known to affect the microstructures of bio-based SMTPUs, a detailed morphological study was carried out using AFM to evaluate the sizes and shape of the hard domains in the various specimens [46]. AFM is also a useful tool for investigation of the physical structure of a sample surface and to quantify the surface roughness. Thus, Figure 4 shows the obtained height and 3D AFM images for the three SMTPU samples, and the corresponding data are presented in Table 5. More specifically, two types of phase contrast can be seen, namely the dark, featureless matrix corresponding to the soft segment, and the bright elements corresponding to the hard segments [47]. Indeed, the high content of the hard segment on the microscale structural level is supported by the presence of numerous spherical globules that correspond to the phase-separated crystalline hard segment domains. As the hard segment content increases upon increasing the [NCO]/[OH] molar ratio, the surface roughness increases, and the phase separation and hard domain structure become more pronounced in the sample morphology. In the case of the MP 1/2.22/1.22 specimen, a sharp hilly surface can be seen, which contains large globules, leading to the highest roughness value of 37.0 nm. The depths and heights of the globules in the z direction on the SMTPU surface were then determined using Nanotec Electronica WSM software (WSM 4.0, 2019), wherein the surface roughness can be observed as a highly spiked region (R_max_ 1204−705 nm) between 31.5 and 37.0 nm. In addition, it was found that the nonuniform distribution of the hard domain at higher contents led to a decrease in surface roughness (i.e., from 54.9 to 39.2 nm), thereby confirming that the size and shape of the hard domain varies depending on the hard segment content and the [NCO]/[OH] molar ratio. More specifically, when the hard segment content increased with variation in the [NCO]/[OH] molar ratio, the hard domains became larger (up to multi-µm in size). As a result, the surface roughness and the degree of phase separation increased, and the hard domain structures became more prominent and visible in the sample morphology. Therefore, these observations support the FTIR results discussed above.

The degrees of crystallinity of the bio-based SMTPUs were then investigated by wide-angle XRD measurements, as outlined in Figure 5. More specifically, it can be seen form this figure that variation in the [NCO]/[OH] molar ratio and hard segment content led to changes in the sample crystallinity. Generally, the peaks observed at 2θ values of 19–23° in the TPU samples correspond to hydrogen bonds between urethane groups in the hard domain [47]. Thus, from the observed diffraction patterns containing relatively broad peaks, it was possible to carry out resolution into two main hard domain regions with maxima at ~19.4 and 23.5°. In the case of a high hard segment content, the most intense peak was observed at 23.5°, and this intensity increased upon increasing the [NCO]/[OH] molar ratio, ultimately indicating the presence of a hard domain containing highly-ordered hydrogen bonding arrangements [19]. Indeed, the phenomenon was observed for the MP 1/2.22/1.22 specimen due to the high weight fraction of hydrogen-bonded urethane groups, as confirmed by FTIR spectroscopy; this high percentage of oriented domains is expected to be advantageous to imparting shape memory properties on the prepared specimens.

As a result, in the case of the MP 1/2.22/1.22 sample, the increment of oriented domains by hydrogen bond formation in the hard segment is expected to be advantageous for the shape memory property.

### 3.3. Thermal Analysis

#### 3.3.1. Differential Scanning Calorimetry (DSC)

The prepared SMTPUs were subjected to DSC, as presented in Figure 6a, wherein several peaks can be observed. More specifically, for the MP 1/2.22/1.22, MP 1/1.09/0.90, and MP 1/1.50/0.50 specimens, the peaks at −31.9, −31.4, and −32.1 °C correspond to the glass transition temperature of the soft segment, the peaks at 58.4, 55.1, and 53.3 °C correspond to the melting temperature of the soft segment, those at 142.9, 138.1, and 130.4 °C correspond to the glass transition temperature of the hard segment, and the peaks at 207.0, 185.1, and 168.3 °C correspond to the melting temperature of the hard segment (see Table 6). Therefore, the ability to detect the individual temperatures for the soft and hard segments may be an indicator of a kinetically favorable and stable phase separation process.

In an SMP, the soft domain is the reversible phase, and is responsible for imparting elasticity and the ability to assume a temporary shape. As reported previously, the shapes of temperature-dependent SMPs can be easily changed above the shape memory transition temperature (T_trans_), and the deformation can be fixed below T_trans_ [2,7,10]. Upon subsequent heating above T_trans_, the original shape can be recovered automatically, as outlined in Figure 7. Since the T_m_ of the soft segment is responsible for determining the shape memory transition temperature, it was considered that the shape memory effects of TPUs could be adjusted to various temperature ranges by cautiously selecting the [NCO]/[OH] molar ratio, thereby bringing about the possibility of expanding their field of application. To examine this possibility in further detail, the MP 1/2.22/1.22 sample was employed as an example SMTPU. This specimen exhibited a relatively sharp endothermic peak at ~60 °C, which was attributed to the T_m_ of the soft segment-associated thermal T_trans_. It was found that, upon increasing the temperature above 60 °C (T_trans_), the switching segments became flexible, and the polymer was deformed elastically. The temporary shape was then fixed by cooling below 60 ℃, and subsequent reheating of the polymer recovered the permanent shape. These results demonstrate the ability of this SMTPU system to exhibit shape memory properties while possessing a high weight fraction of hydrogen-bonded urethane groups (*X_b_*).

As previously reported, the hard domain is responsible for the material properties at high temperatures, wherein multiple transitions can take place depending on the content of the hard segment present in the matrix [21]. In addition, the different endotherms observed at high temperatures have been the subject of much controversy over the past decades [45,46], For example, Seymour and Cooper [48] described that multiple endotherms could represent decomposition of the short-range ordered hard segment domains (hard segment T_m_ 1) and long-range ordered hard segment domains (hard segment T_m_ 2). In our systems, it was found that these two temperatures which were observed for melting of the hard segments shifted toward higher temperatures as the content of hard segment increased, as outlined in Table 6. The different transitions associated with relaxation of the hard phase polymer chains, the inter-mediate temperatures related to microphase mixing of the hard and soft segments, and the two-step melting process corresponding to an ordered structure present in the hard phase (TI, TII) have also been reported [49,50]. More specifically, previous works have suggested that the presence of double or triple melting temperatures indicates the presence of a higher-ordered hard phase and can be a consequence of a higher phase separation [49]. Indeed, the MP 1/2.22/1.22 specimen exhibited three T_m_ values for the hard segment as a result of superior hard segment ordering and the formation of stronger and more stable hard domains; these results are supported by the FTIR, XRD, and AFM experiments described above.

#### 3.3.2. Dynamic Mechanical Analysis (DMA)

The dynamic mechanical properties of the prepared specimens are presented in Figure 6b and Table 7, wherein the relationship between the storage modulus (G′) and the temperature can be seen, along with that between the tan δ value and the temperature. These plots reveal the various thermal transitions taking place in the samples and indicate several useful temperature ranges. More specifically, below a temperature of −40 °C, the G′ values of the three specimens remain relatively constant due to restricted molecular motion (i.e., vibrations and short-range rotations of the soft segment) at such low temperatures. Therefore, it was expected that the MP 1/2.22/1.22 specimen would exhibit superior shape fixation properties because of its higher modulus in the glassy state as in [51]. Upon cooling further to −50 °C, the value of G′ gradually decreased, corresponding to the T_g_ of the soft segment. Beyond this point, the rubbery plateau is reached, wherein the storage modulus increases with an increase in the hard segment content due to the presence of a more crosslinked structure, structural reinforcement, and a high elastic recovery at high temperatures [2]. Considering that the storage modulus defines the energy stored elastically by a material at deformation and supplies information regarding the polymer stiffness [38], the increased storage modulus observed at higher contents of the hard segment can be attributed to the presence of a glassy state due to the crystallinity of this domain. As shown in Figure 6b, the highest values of G′ were within the range of 3.3–3.5 MPa. In addition, the relative crosslinking densities of the polyurethane species were associated with the observed storage moduli in the rubbery plateau region, wherein a higher crosslinking density was associated with a greater hard segment content, indicating that the chain segments were more closely restricted by net points to inhibit their mobility. Due to the fact that shape recovery in thermoset polymers is dependent on the covalently-bonded net points [10], it was, therefore, considered that polymers with greater hard segment contents would exhibit enhanced shape recovery properties.

The flex temperature (T_flex_), which is also provided in Table 7 for the three SMTPU specimens, is defined as the temperature at the start of the rubbery plateau region, while T_flow_ corresponds to the point at which melting of the hard segment takes place. Thus, based on the results presented in Table 7, it appears that the increase in T_flow_ may be the result of increased hydrogen bonding in the hard domain [35]. In addition, the modulus at the rubbery plateau, i.e., in the T_flex_−T_flow_, is a function of the crosslinking density in the hard segment and reinforcement by the separated hard domain. Importantly, it has been reported that TPUs based on segmented copolymers can be synthesized with a range of rubbery moduli via variation in the hard segment content [37]. Upon considering our prepared specimens, it can be seen that the rubbery plateau in the MP 1/2.22/1.22 is extended compared to that of the MP 1/1.50/1.50 sample, with a higher rubbery modulus also being observed for the former. Furthermore, as suggested above, the relatively stable value of the storage modulus at low temperatures indicates that that no phase transitions occur within this temperature range due to effective phase separation. These results are confirmed by the higher *X_b_* obtained for the MP 1/2.22/1.22 specimen following FTIR analysis. Thus, considering the DSC and DMA results, a temperature range of 60–70 °C was selected as the transition temperature to assess the shape-memory properties of the SMTPUs.

#### 3.3.3. Thermogravimetric Analysis (TGA)

The thermal stabilities of bio-based SMTPUs were evaluated by TGA. The resulting mass losses and derivative curves can be seen in Figure 6c, while the corresponding data for the degradation parameters are summarized in Table 8. More specifically, the mass losses curves of the SMTPUs indicate two decomposition stages, namely a stage at 330–332 °C corresponding to decomposition of the hard segment, and a second stage at 421–423 °C related to decomposition of the soft segment [45]. In addition, the first degradation temperature (T_1st_) was found to correlate with the content of the hard segment, wherein the higher thermal stability observed for the MP 1/2.22/1.22 can be attributed to the more tightly packed structure and the enhanced crystallinity of this dominant domain [48], as confirmed by FTIR, DSC, and AFM examinations. The second degradation step was associated with breakage of the ester bonds within the PPS polyol of SMTPU. In this case, the MP 1/1.50/0.50 exhibited a maximum rate of decomposition at ~420 °C, thereby indicating its superior thermal stability compared to conventional polyols [48]. The initial degradation temperature of the second stage (T_2nd_) was found to be similar for all samples. However, the bio-based polyol soft segment underwent more facile degradation when lower hard segment contents were present in the samples, thereby influencing the third stage (T_3rd_) of SMTPU degradation. More specifically, this third stage is related to degradation of the remaining structure and of any residues produced during the second stage of degradation [52,53].

### 3.4. Mechanical Properties

Figure 8a and Table 9 provide the mechanical properties (initial modulus, tensile strength, elongation, and hardness) for the prepared SMTPU specimens at room temperature. In this context, it should be noted that the area under the stress–strain curve increases with increases in the tensile strength and elongation at the break; this region is important in SMPs because it represents the stored strain energy during stretching and drives the strain recovery upon the release of stress in the elastomer’s rubbery state. Based on the obtained results, it was apparent that the mechanical performances of the TPUs were closely related to their compositions; in general, the physical crosslinking present in the hard segment allowed it to act as a reinforcing unit, while the soft segment is responsible for the material flexibility due to the presence of long linear polyol chains. From the data presented in Table 7, it is clear that higher [NCO]/[OH] molar ratios led to greater hard segment contents in the bio-based SMTPUs. As a result, an increase in the hard segment content from 30 to 40 wt% resulted in a corresponding increase in the tensile strength from 23.9 to 27.3 MPa. In addition, the MP 1/2.22/1.22 specimen displayed a high elongation at break (358%), along with a tensile strength >27 MPa, and a higher hardness (Shore A) value. Furthermore, it was found that the elongation at break decreased with an increase in the hard segment content. Since the hardness of a material is closely connected with the crosslinking density in the hard segment, a higher crosslinking density leads to a reduced elasticity and a more rigid material. It was found that the samples prepared using the bio-based polyol exhibited excellent hardness values ranging from 84 to 92 Shore A.

To determine the suitability of an SMTPU for use in elastomer applications, it is necessary to assess its material behavior under repeated loading conditions [5,7,11]. Thus, the prepared specimens were subjected to cyclic tensile tests, as outlined in Figure 8. More specifically, stress–strain hysteresis curves were recorded for the SMTPUs under constant-strain cyclic loading at 25 and 75 °C to investigate the compressive behaviors and recoverability of the three specimens containing different hard segment contents. The temperature selected for these tests was based on the value of T_trans_ determined by DSC and DMA, since this temperature was associated with melting of the soft segment (i.e., the recovery temperature); measurements were also carried out at 25 °C for comparison. Figure 8b,c shows the evolution of the stress–strain curves over 10 cycles at a maximum strain of ~100%, wherein it can be seen that the SMTPU specimens exhibit the typical flag-shaped hysteresis. More specifically, it was observed that all SMTPUs exhibited a virtually constant level of recovery after the first cycle, in which hysteresis was observed and the residual strain gradually increased. This behavior is typical of thermoplastic elastomers [52,53] and can be seen as the adopting of a largely constant structure after an initial change in shape with the alignment of the polymer microstructure [54]. It should be noted that, at both temperatures, the MP 1/2.22/1.22 specimen exhibited a cyclic hysteresis that was more pronounced in the first cycle, and then gradually decreased in the following cycles. In addition, the slope of the hysteresis loop was found to decrease gradually with a decreasing hard segment content, which correlates with the dynamic stiffness of the material. Furthermore, the slope variation in the hysteresis loop corresponded to the storage modulus as a function of the dynamic strain amplitude results presented in Figure 6b, which indicates that the storage modulus decreases with an increasing content of the soft segment. In the case of the MP 1/1.50/0.50 specimen, a lower hysteresis and cyclic stress value were obtained compared to the MP 1/2.22/1.22 sample, and the area of the hysteresis loop became smaller upon increasing the number of cycles. Moreover, the change in the hysteresis loop was more significant in the early cycles, and the hysteresis loop tended to stabilize at a higher number of cycles. It was also found that the decrease in the transformation stress was more pronounced at 25 °C as the content of the hard segment was in-creased; therefore, the area of the hysteresis loop became smaller during cycling at 75 °C. It can also be seen in Figure 8 that the accumulated residual strain led to a shift of the hysteresis loops to the right upon increasing the number of cycles; the maximum residual strain was ~20%. The reduced increment of the residual strain during cyclical loading can therefore be accounted for by the accumulation of permanent localized deformations in the early cycles, and their subsequent disappearance during later cycles. Under cyclic loading, the energy dissipated per cycle gradually decreases as a result of narrowing the hysteresis loop. For thermoplastic elastomers, it is generally observed that the shape recovery behavior in the first two cycles is different from that in the later cycles; this can be attributed to distribution of the crystalline phase, in addition to the thermomechanical history of chain formation [55]. Moreover, as the number of fatigue cycles increases, the equivalent viscous damping also decreases rapidly. However, the rate of decrease of the dissipated energy and the equivalent viscous damping tends to be reduced as the number of cycles increases. Overall, the stress–strain curves exhibited highly recoverable and repeatable behaviors, thereby indicating that the SMTPUs could potentially retain their elastomeric properties even after the construction of complexes by FDM 3D printing. Indeed, such reproducibility is of particular importance when considering the long-term performance of an SMTPU.

### 3.5. Fabrication of the Bio-Based SMTPU Filaments

Figure 9a shows photographic images of the SMTPU filaments observed by optical microscopy. These images were recorded using various filament diameters (1.55–1.80 mm), which were obtained by adjusting the screw speed and screw temperature. As shown, the cross-sectional and surface images indicate the presence of a co-continuous and even surface, in addition to a good miscibility. More specifically, the surface images appear smooth, and the cross-sectional images show uniform white filaments. In addition, as presented in Figure 9b, the tensile modulus was found to gradually increase from 20 to 31 MPa upon increasing the content of the hard segment from 30 to 40 wt%. Furthermore, the elongation at break values of the MP 1/1.22/1.22, MP 1/1.90/0.90, and MP 1/1.50/0.50 specimens were determined to be 605, 807, and 1134%, respectively, while their hardness values ranged from 80 to 87 Shore A.

### 3.6. Shape Recovery Properties

#### 3.6.1. Shape Fixing and Recovery Properties of the Bio-Based SMTPU Films

The shape memory effects of the prepared SMTPU films were measured using SMP program Ⅰ with T_trans_ set at 70 °C (i.e., corresponding to the T_m_ of the soft segment, as determined by DSC). Initially, the flat specimens were subjected to shape fixation by helical deformation at 70 °C over a period of 30 min and subsequent cooling to room temperature (20 °C). After subsequent heating to 75, 80, or 85 °C, the initial shape was recovered; this shape memory process can be seen in Figure 10a. It was found that all bio-based SMTPU films underwent full fixation to form a helical shape; however, they later recovered quickly to their original straight shapes. More specifically, the MP 1/1.22/1.22 specimen exhibited rapid shape recovery at 85 °C, re-forming the flat specimen within 190 s. Therefore, these results demonstrate that the bio-based SMTPU films were able to recover their initial shapes more rapidly compared to previously reported SMTPUs [19]. As previously reported, in shape memory TPUs, the hard segment is responsible for shape recovery, while the soft segment phase is responsible for shape fixing [56]. Therefore, it was considered that the prepared bio-based SMTPUs exhibited excellent shape fixing and shape recovery properties due to their high rubbery moduli and slow chain relaxation. Subsequently, the shape recovery ratios (R_r_) were obtained using SMP program Ⅱ to assess the quality of the SMPs. As presented in Figure 10b, the prepared specimens exhibited R_r_ values between 96 and 100%, thereby demonstrating their excellent ability to fix a temporary shape and recover their permanent shape. In particular, the MP 1/2.22/1.22 specimen exhibited the fastest shape recovery due to its high hard segment content, which leads to stable shape memory properties due to chain mobility restrictions from the low levels of the PPS polyol soft segment. According to the obtained results, the shape recovery ratio increases upon increasing the recovery temperature and this was likely due to the fact that a lower recovery temperature leads to a reduced mobility in the hard segment chains, ultimately resulting in a longer relaxation. As discussed in the previous section, the large areas under the stress–strain curves of the pre-pared bio-based SMTPUs indicate their ability to store relatively large amounts of energy in their temporary shapes. An improvement in the storage modulus would also be expected to increase the shape recovery ratios of the samples [57]. Based on the above results, shape recovery ratios of ~100% would be expected upon the application of these bio-based SMTPUs to 4D printing.

#### 3.6.2. Shape Fixing and Recovery of Bio-Based 3D-Printed SMTPUs

As previously reported, 3D-printed auxetic re-entrants possess strong potential applications in the design of energy absorption foams [58], smart composites [59], metamaterials [60], sensors and actuators [61], and biomedical materials and devices [62]. In addition, such materials could be used to obtain smart mechanical metamaterials that can be responsive to external loads and/or environmental conditions, such as temperature, light, and humidity. Ultimately, the application of these materials could lead to new systems, sensors, and/or actuators with extensive engineering functions. Prior to carrying out the 3D printing process, a preliminary test was conducted (see Appendix A) to optimize the printing and loading temperatures. Appendix A shows sample images of the 3D-printed SMTPUs prepared from the MP 1/2.22/1.22, MP 1/1.90/0.90, and MP 1/1.50/0.50 specimens. Thus, to confirm the suitability of the prepared samples for use in 4D printing, the shape memory behaviors of the 3D-printed auxetic re-entrant were tested using SMP program Ⅰ. Upon evaluation of the recovery temperature and the unfolding angle over time (Figure 11b), it was found that an increase in the recovery temperature led to a shorter recovery time (i.e., the time required for the sample with a high hard segment content to reach the final curvature (0°)); the time required to reach 90° of curvature from a circular-like shape showed a similar trend. Initially, the 3D-printed auxetic re-entrant of specimen MP 1/2.22/1.22 was fixed in a circular-like shape, and it was found to undergo excellent shape recovery over 55 s at 90 °C (see Figure 11a). Similarly, recovery times of 60 and 90 s were obtained at this temperature for the MP 1/1.90/0.90 and MP 1/1.50/0.50 specimens. As shown in Figure 11, all samples reached 100% recovery relatively rapidly upon heating at 80–90 °C, thereby confirming their excellent shape recovery properties, which were most apparent for the MP 1/2.22/1.22 specimen. Overall, the various analytical results presented for this specimen indicate its potential for use as a shape memory material.

In particular, MP 1/2.22/1.22 showed good shape recovery and good shape fixing performances. As we discussed above, the results of FTIR, AFM, DMA, etc., exhibited an increment of oriented domains by hydrogen bond formation in the hard segment with a high weight fraction of hydrogen-bonded urethane groups (*X_b_* = 0.92), the largest hilly surface containing large globules, and the highest roughness value of 37.0 nm as a result of the hard domain. Further, in DSC and DMA, it presented multiple melting processes, accordant with an ordered structure in the hard phase and extended rubbery plateau, and higher rubbery modulus, respectively. Based on these results, it was expected that it would be advantageous for shape memory properties, and the results reflected excellent shape memory properties.

## 4. Conclusions

Recently, shape memory polymers have received extensive attention due to their unique deformation mechanisms and various shape-changing behaviors. To meet these growing demands on renewable sources, this study aimed to synthesize a novel shape memory thermoplastic polyurethane (SMTPU) elastomer from bio-based materials. For this purpose, bio-based polyol, 1,3-propanediol, and aromatic diisocyanate were combined via the pre-polymer method with bio contents of 66−73%. Upon the investigation of various parameters on the shape memory characteristics of the obtained specimens, it was found that all bio-based SMTPUs were characterized by a high weights fraction (0.77−0.92) of hydrogen-bonded urethane groups (*X_b_*), independent of the hard segment content present in the sample. In addition, only ~0.05–0.99 wt% of the hard segment was found in the soft phase (*W′*_2_), demonstrating a high degree of phase separation in all samples, although this was particularly apparent for the MP 1/2.22/1.22 specimen. Following observation of the prepared SMTPUs by atomic force microscopy, it was found that, when the content of the hard segment was increased, the hard segment domains became larger. As a result, the surface roughness and the extent of phase separation increased, leading to the hard domain structures becoming more pronounced and visible in the sample morphology. Furthermore, following evaluation by differential scanning calorimetry and dynamic mechanical analysis, the relatively-constant storage modulus indicated that no phase transitions occurred within the examined temperature range due to effective phase separation. Based on these results, a transition temperature range of 60–70 °C was selected to assess the shape memory properties of the specimens. In terms of the mechanical properties, an increase in the content of the hard segment from 30 to 40 wt% resulted in a corresponding increase in tensile strength from 23.9 to 27.3 MPa, which was accompanied by a decrease in the elongation at break from 552 to 358% and an increase in the hardness value from 82 to 92 Shore A. Subsequently, cyclic tensile tests confirmed the reproducibility of the long-term performances of the SMTPUs, since the maintenance of elastomeric properties is of particular importance for a material even after three-dimensional (3D) printing. Moreover, shape memory analysis demonstrated that the bio-based SMTPU films and 3D-printed auxetic RE became completely fixed, and, following subsequent reheating, they quickly re-covered their original shapes (recovery ratios = 100%). These results indicated that the prepared samples were not only able to fully recover their initial shapes, but that their recovery speeds were significantly shorter than those of previously reported SMTPUs. In the future, it is expected that this strategy will be extended to advanced 3D/4D printing technologies, in addition to being employed in the preparation of rapid and sophisticated reversible shape memory devices.

## Figures and Tables

**Figure 1 materials-16-01072-f001:**
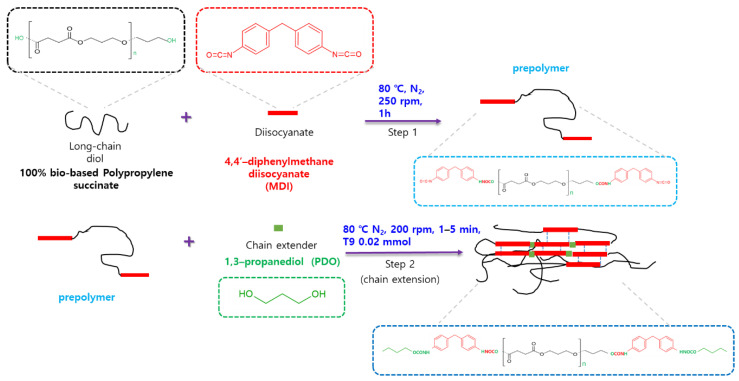
Synthetic scheme outlining preparation of the bio-based SMTPU via the two-step prepolymer process.

**Figure 2 materials-16-01072-f002:**
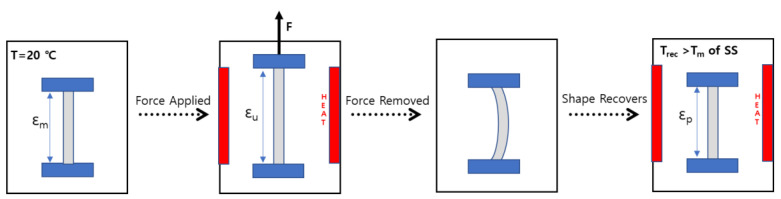
Schematic of the shape memory test carried out for the bio-based SMTPU films.

**Figure 3 materials-16-01072-f003:**
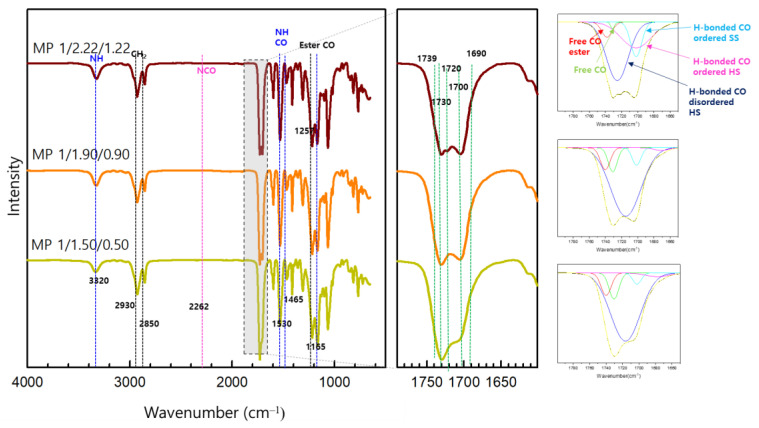
FTIR spectra of the bio-based shape memory thermoplastic polyurethanes, and curve fitting of the carbonyl region.

**Figure 4 materials-16-01072-f004:**
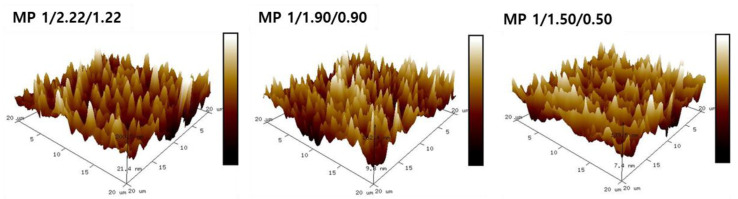
AFM phase images of the freeze-fractured surfaces of the prepared bio-based SMTPUs.

**Figure 5 materials-16-01072-f005:**
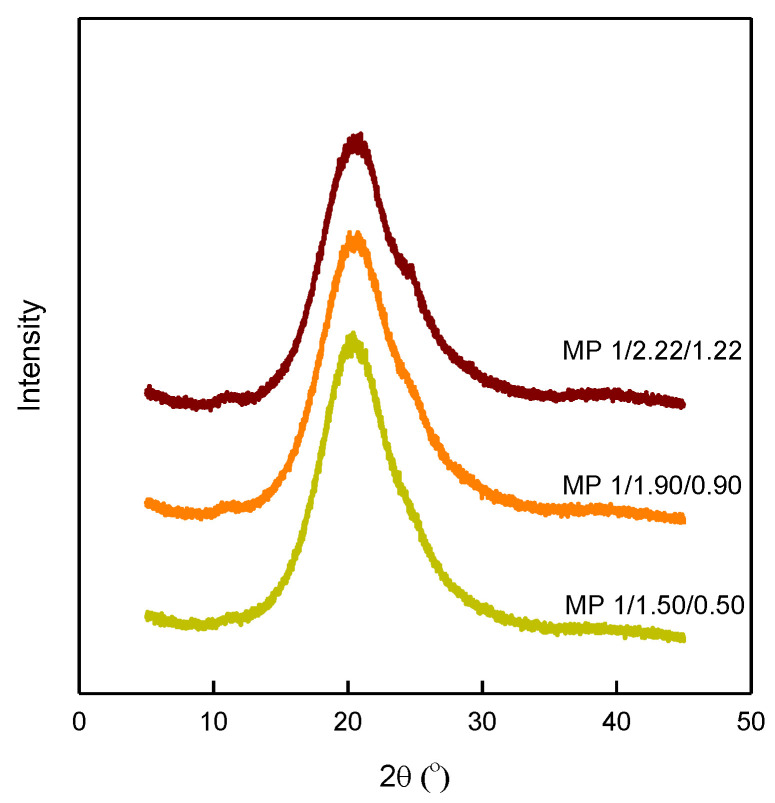
X-ray diffractograms of the bio-based SMTPUs.

**Figure 6 materials-16-01072-f006:**
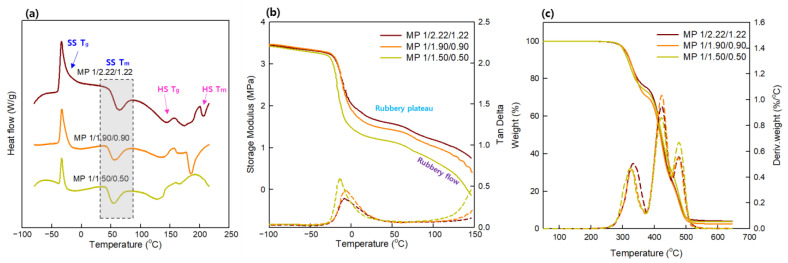
(**a**) Differential scanning calorimetry results for the first heating cycle. (**b**) Plots of the storage modulus versus temperature and tan δ versus temperature from dynamic mechanical analysis. (**c**) TGA curves and their derivative curves for the prepared bio-based SMTPUs.

**Figure 7 materials-16-01072-f007:**
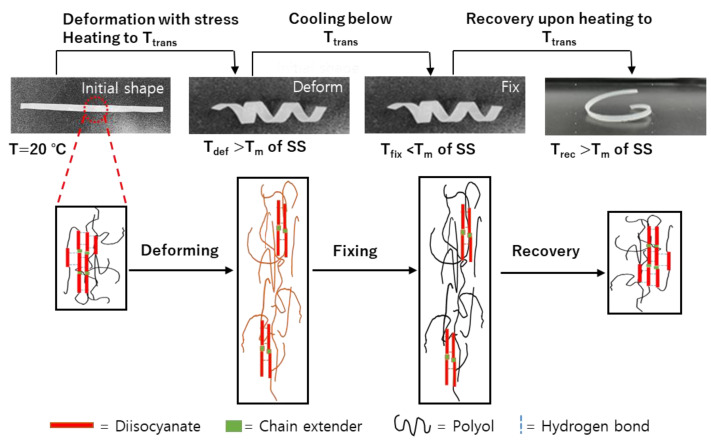
Shape memory cycle of the bio-based shape memory thermoplastic polyurethanes.

**Figure 8 materials-16-01072-f008:**
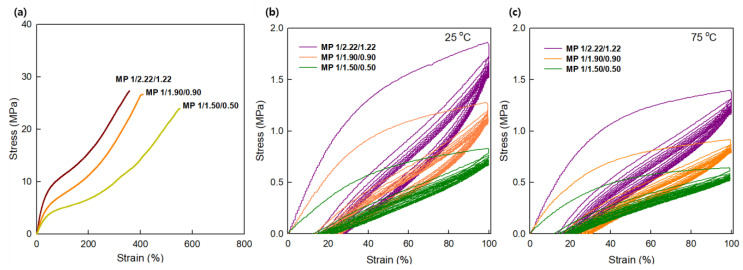
(**a**) Tensile stress–strain curves of the bio-based SMTPUs. Stress–strain hysteresis curves obtained during cyclic testing (applied strain = 100%) at (**b**) 25 °C and (**c**) 75 °C.

**Figure 9 materials-16-01072-f009:**
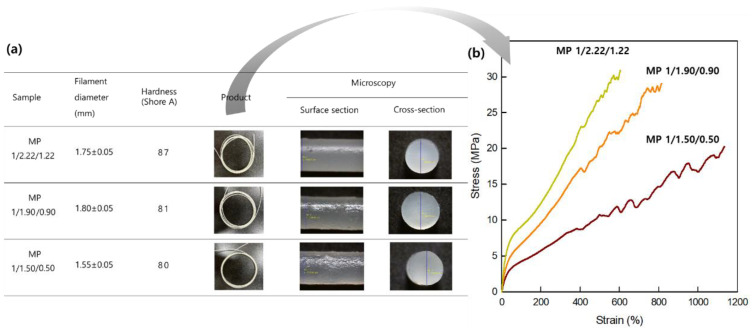
(**a**) Parameters and surface and cross-sectional images for the SMTPU filaments. (**b**) Tensile test curves for the prepared SMTPU filaments.

**Figure 10 materials-16-01072-f010:**
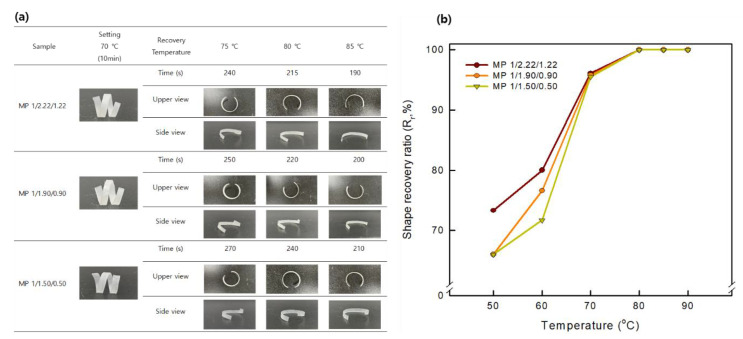
(**a**) Photographic images of the shape memory behaviors of the prepared SMTPUs, and (**b**) their corresponding recovery ratios (Rr) at different reheating temperatures.

**Figure 11 materials-16-01072-f011:**
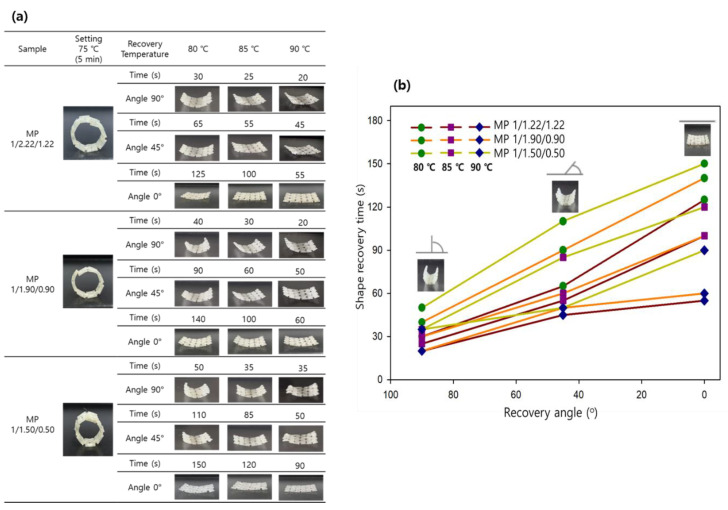
(**a**) Photographic images showing the shape memory behaviors of the auxetic re-entrant SMTPU samples subjected to different recovery temperatures. (**b**) Plot of the shape recovery time versus the recovery angle under different recovery temperature conditions (i.e., 80, 85, and 90 °C) for the 3D-printed re-entrants.

**Table 1 materials-16-01072-t001:** Formulation of the bio-based thermoplastic polyurethane samples.

Sample	NCO/OH	HS Content (wt%) ^a^	Content of Bio-Based Sources (wt%) ^b^
MP 1/2.22/1.22	1:1	39.3	66.3
MP 1/1.90/0.90		35.2	69.2
MP 1/1.50/0.50		29.2	73.4

^a^ HS: Hard segment content is defined as the ratio of the mass of diisocyanate and chain extender components to the total mass. ^b^ Bio-content is defined as the ratio of the mass of bio-based components to the total mass.

**Table 2 materials-16-01072-t002:** Melt-extrusion conditions used to fabricate the SMTPU filament from TPU films.

Sample	Screw Temperature (°C)	Screw Speed (rpm)	Winding Position (cm)	Nozzle Diameter (mm)
MP 1/2.22/1.22	207	13.2	90	1.5
MP 1/1.90/0.90	205	13.0	90	1.5
MP 1/1.50/0.50	200	13.2	90	1.5

**Table 3 materials-16-01072-t003:** Percentages of the various C=O species present in the shape memory thermoplastic polyurethanes, as determined by deconvolution of the FTIR absorbance bands in the range of 1750–1680 cm^−1.^

Sample	Free C=O Ester in Soft Domain (%) 1739 cm^−1^	Free C=O in Hard Domain (%) 1730 cm^−1^	H-Bonded C=O in Disordered Hard Domain (%) 1720 cm^−1^	H-Bonded C=O in Ordered Hard Domain (%) 1700 cm^−1^	H-Bonded C=O in Disordered Soft Domain (%) 1690 cm^−1^	*X_b_*
MP 1/2.22/1.22	13.5	5.0	28.5	39.0	14.2	0.92
MP 1/1.90/0.90	13.8	11.6	35.8	18.7	20.2	0.80
MP 1/1.50/0.50	12.4	11.9	36.2	10.5	29.0	0.77

**Table 4 materials-16-01072-t004:** Calculated terms related to microphase separation in the bio-based SMTPUs.

Items	*f*	*X_b_*	*W′* _2_	MP	SP	HP
Sample						
MP 1/2.22/1.22	0.39	0.92	0.05	0.02	0.63	0.37
MP 1/1.90/0.90	0.35	0.80	0.10	0.03	0.68	0.32
MP 1/1.50/0.50	0.29	0.77	0.99	0.03	0.74	0.26

**Table 5 materials-16-01072-t005:** Surface characterization of the AFM phase images of the SMTPU films.

Sample	Surface Area (µm^2^)	R_q_ (nm) ^a^	R_a_ (nm) ^b^	R_max_ (nm) ^c^
MP 1/2.22/1.22	409	54.9	37.0	1204
MP 1/1.90/0.90	405	44.3	34.9	670
MP 1/1.50/0.50	404	39.2	31.5	705

Surface area: the total area of examined sample surface. ^a^ R_q_ (rms): Standard deviation of the Z values within the given area. ^b^ R_a_ (mean roughness): Mean value of the surface relative to the center place. ^c^ R_max_ (max height): Difference in height between the highest and lowest points on the surface relative to the mean plane.

**Table 6 materials-16-01072-t006:** Summary of the DSC results for the prepared bio-based SMTPUs.

Sample	SS T_g_ (°C)	SS T_m_ (°C)	HS T_g_ (°C)	HS T_m_ 1 (°C)	HS T_m_ 2 (°C)
MP 1/2.22/1.22	−31.9	58.4	142.9	181.0	207.0
MP 1/1.90/0.90	−31.4	55.1	138.1	161.6	185.1
MP 1/1.50/0.50	−32.1	53.3	130.4	168.3	-

**Table 7 materials-16-01072-t007:** Summary of the dynamic mechanical analysis results for the bio-based SMTPUs.

Sample	T_g_ (°C)	G′_25_ (MPa) ^a^	T_flex_ (°C) ^b^	T_flow_ (°C) ^c^
MP 1/2.22/1.22	−6.7	1.69	16.6	126.7
MP 1/1.90/0.90	−7.5	1.53	14.5	104.9
MP 1/1.50/0.50	−13.9	1.23	10.2	69.0

^a^ G′_25_: Storage modulus at 25 °C. ^b^ T_flex_: The flex temperature (temperature at the start of the rubber plateau region). ^c^ T_flow_: The flow temperature (the temperature where the storage modulus G′ reached 1 MPa).

**Table 8 materials-16-01072-t008:** Thermal decomposition characteristics of the obtained bio-based SMTPUs.

Sample	Initial Degradation Temperature for Each Step		
	T_1st_ (°C)	T_2nd_ (°C)	T_3rd_ (°C)
MP 1/2.22/1.22	331.8	421.1	475.0
MP 1/1.90/0.90	330.5	421.2	477.0
MP 1/1.50/0.50	322.2	422.7	479.2

**Table 9 materials-16-01072-t009:** Mechanical properties and hardness values of the bio-based SMTPUs.

Sample	Initial Modulus (MPa)	Tensile Strength (MPa)	Elongation (%)	Energy (J)	Hardness (Shore A)
MP 1/2.22/1.22	28.66 ± 0.3	27.34 ± 0.2	358.26 ± 4.5	0.70	92
MP 1/1.90/0.90	17.11 ± 0.15	26.64 ± 0.2	409.43 ± 4.0	0.82	86
MP 1/1.50/0.50	9.67 ± 0.1	23.93 ± 0.2	551.59 ± 5.4	0.66	84

## Data Availability

The data presented in this study are available on request from the corresponding author.

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
