# Peer review of "Synthesis of Novel Shape Memory Thermoplastic Polyurethanes (SMTPUs) from Bio-Based Materials for Application in 3D/4D Printing Filaments"

_materials, 2023, doi:10.3390/ma16031072_

Round 1
Reviewer 1 Report
Please find attached file

Author Response
Comment 1: In the title, I will recommend adding in 4D/3D Printing Filaments, as it is said in the conclusion
→ Thank you for your valuable comment. We have made editorial changes as suggested.
“Synthesis of Novel Shape Memory Thermoplastic Polyurethanes (SMTPUs) from Bio-based Materials for Application in 3D/4D Printing Filaments”
Comment 2: In the introduction, it should be mentioned if the TPUs have been used in 4D printing applications. Please add the relevant literature on the use of TPU in 4D printing. If this is the first time that a study like that has been done, the authors can specify that it is the first time the TPU has been assessed for 4 D application
→ We appreciate the suggestions. We have made editorial changes as suggested.
“TPUs can be easily processed for use in casting, injection-molding, and extruded filaments for 3D/4D printing [28], which is of particular interest due to the rapid development of the 3D/4D printing industry. Among the 3D/4D printing materials reported to date, the TPU filaments have been reported to exhibit an excellent elasticity and flexibility, in addition to good shock absorption properties. TPU materials have been used previously in energy absorption, soft-robotics, drug delivering devices, biomedical devices, tissue engineering, mechanical actuator and prosthetics application [24, 27–32]. For instance, Bates et al. [27] developed a parametric study that undertaken, capturing the energy absorbing capability of hexagonal arrays manufactured from two types of TPU, with relative densities 0.18–0.49. Yap et al. [28] presents a novel technique for direct 3D printing of soft pneumatic actuators using 3D printers based on fused deposition modeling (FDM) technology with thermoplastic elastomer filament (shore hardness of 85A). More recently, Rahmatabadi’ group [32] has proposed the combination of the microscopic concept of SMPs and multimaterial printing of a
thermoplastic elastomer with FDM is introduced to generate a novel 4D printing method without SMP and the extra operation such as synthesis and blending. In another study, Kızıltay et al. [31] synthesized thermoplastic poly(ester-urethane) (PEU) was from L-lysine diisocyanate (LDI) and polycaprolactone diol (PCL) for tissue engineering applications.”
- Bates, S.R.; Farrow, I.R.; Trask R.S. 3D printed polyurethane honeycombs for repeated tailored energy absorption. Mater. Des. 2016,112, 172–183.
- Yap, H.K.; Ng, H.Y.; Yeow C.-H. High-force soft printable pneumatics for soft robotic applications. Soft Rob., 2016, 3(3), 144-158.
- Jia, H.; Huang, ZZ.; Fei, P.J.; Dyson, Z.; Zheng, X. Wang Bilayered polyurethane/dipole-dipole and H-bonding interaction reinforced hydrogels as thermo-responsive soft manipulators. J. Mater. Chem. B 2017, 5(41), 8193-8199.
- Ho, E.; Chen, Y.; Traore, Y.; Li, A.; Fowke, K. Development of polyether urethane intravaginal rings for the sustained delivery of hydroxychloroquine. Drug Des. Devel. Ther. 2014, 1801.
- Kızıltay A, Fernandez AM, Roman JS, Hasirci V, Hasirci N. Lysine Based Poly(ester-urethane) Films for Tissue Engineering Applications. J Biomater Tissue Eng 2012, 2, 143–53.
- Rahmatabadi, D.; Aberoumand, M.; Soltanmohammadi, K.; Soleyman, E.; Ghasemi, I.; Baniassadi, M.; Baghani, M. 4D Printing‐Encapsulated Polycaprolactone–Thermoplastic Polyurethane with High Shape Memory Performances. Advanced Engineering Materials, 2022, 2201309.
Comment 3: In the materials section, how do the authors know that 1,3-propanediol was bio-based? Is there a certification from Sigma-Aldrich? The author should explain how they found it..
→ We appreciate this comment. We checked below sentence again. Product name of 1,3-propanediol from Dupont Tate & Lyle bioproducts is SUSTERRA (R) PROPANEDIOL.
“2.1. Materials
The hard segment was prepared using MDI and bio-based 1,3-propanediol (PDO, chain extender). These two components were purchased from Sigma Aldrich (Burlington, MA, USA), and Dupont Tate & Lyle BioProducts (SUSTERRA (R) PROPANEDIOL ,Wilmington, DE, USA), respectively.”
Comment 4: In section 2.4, the author should explain why they choose to print a film and not a more complex shape to investigate Shape recovery properties.
→ We appreciate your valuable comment.
First, in order to simply test the shape recovery performance, a film was manufactured using a press in the laboratory. Even if a more complicated shape was not confirmed, it was necessary to check the shape memory performance once again by printing with a auxetic re-entrant structure in 3D printing, so it was necessary to check it in a simple film form.
Comment 5: In equation 2, the f definition should be added in the text
→Thank you for your valuable comment. We added f definition in the text.
“Eqs. (2)−(5) (f: the weight fraction of hard segment before synthetic process) [43]:”
Comment 6: In section 3.3.1, the melting temperature for the hard segment for MP1/1.5/0.5 is said at 168.3 °C. However, in Figure 6 a, the peak for that temperature doesn't show. Furthermore, that temperature is not reported in table 6. Could you explain how do you found that value? Why didn't you report on Table 6?
→ We appreciate your valuable indication.
We regret that in the case of samples MP1/1.5/0.5, the melting peak was small and difficult to distinguish visually. Please check again with the big picture. What you think is missing from the table is that there are several melting peaks of the hard segment of other samples, so it is divided into 1 and 2, so in the case of MP1/1.5/0.5, there is only one peak, so it seems to be missing.
Comment 7: In figure 6 b, it should be pointed out that the data obtained by the DMA analysis
→ Thank you for your valuable comment. We have made editorial changes.
“Figure 6. (a) Differential scanning calorimetry results for the first heating cycle. (b) Plots of the storage modulus versus temperature and tan δ versus temperature from dynamic mechanical analysis. (c) TGA curves and their derivative curves for the prepared bio-based SMTPUs.”
Comment 8: In table 7, please add the description of symbols on the table in the table's label.
→ Thank you for your valuable comment. We have made editorial changes as suggested.
“ a G′25 : Storage modulus at 25 °C.
b Tflex: The flex temperature (temperature at the start of the rubber plateau region)
c Tflow: The flow temperature (the temperature where the storage modulus G′ reached 1 MPa).”
Comment 9: In section 3.3.3, replay figure 5c by Figure 6c
→ Thank you for your valuable comment. We have made editorial changes as suggested.
“The thermal stabilities of bio-based SMTPUs were evaluated by TGA. The resulting mass losses and derivative curves can be seen in Figure 6c,”

Reviewer 2 Report
1. The authors should explain the novelty of their work.
2. References in the manuscript can be reduced to 50 or 60
3. References should not come from a particular country. Equal weightage to be given to authors from all countries. Please provide the number of references from each country in the rebuttal.
4. Conclusions are very lengthy. They need to be crisp and point wise
5. . Furthermore, following evaluation by differential scanning calorimetry and dynamic mechanical analysis, the relatively constant storage modulus indicated that no phase transitions occurred
within the examined temperature range due to effective phase separation.
Explain how the authors arrived at this conclusion?
6. Moreover, shape memory analysis demonstrated that the bio-based SMTPU films became completely fixed in the form of a helical shape, and following subsequent reheating, they quickly recovered their original straight shapes. What is the reason for this behaviour?
7. Figure 8. please make the legends and markings readable. It is too small.
8. In addition, the relative crosslinking densities of the polyurethane species
were as-sociated the observed storage moduli in the rubbery plateau region, wherein a
higher crosslinking density was associated with a greater hard segment content, indicating that the chain segments were more closely restricted by net points to inhibit their mobility
How is the above statement confirmed experimentally?
Author Response
Comment 1: The authors should explain the novelty of their work.
→ Thank you for your valuable comment. We have explained the novelty of this paper. Relevant information is described in section of conclusion.
-Application and synthesize a novel shape memory thermoplastic polyurethane (SMTPU) elastomer from bio-based materials. 100% bio-based polyol, polypropylene succinate (PPS), aromatic diisocyanate (MDI), and bio-based 1,3-propanediol (PDO, chain extender) were combined via the prepolymer method with bio-based source contents of 66−73%.
-Fabrication of bio-based SMTPU filaments for 4D printing by melt extrusion.
-The development of eco-friendly temperature-dependent bio-based SMTPUs for 4D printing with excellent shape recovery
“4. Conclusions
this study aimed to synthesize a novel shape memory thermoplastic polyurethane (SMTPU) elastomer from bio-based materials. For this purpose, 100% bio-based polyol, and 1,3-propanediol, and aromatic diisocyanate were combined via the prepolymer method to give bio-based SMTPUs with bio contents of 66−73%. notably, bio-based SMTPU filaments for 3D printing were successfully by melt extrusion.
notably, bio-based SMTPU filaments for 3D printing were successfully by melt extrusion.
Moreover, shape memory analysis demonstrated that the bio-based SMTPU films became completely fixed in the form of a helical shape, and following subsequent reheating, they quickly re-covered their original straight shapes (recovery ratios = 100%).”
Comment 2: References in the manuscript can be reduced to 50 or 60
→ We appreciate the suggestions. We have made editorial changes. We tried to reduce the number of references to 62 as much as possible.
Comment 3 References should not come from a particular country. Equal weightage to be given to authors from all countries. Please provide the number of references from each country in the rebuttal.
→ We appreciate the comment. We provided the number of references from each country in the rebuttal.
Comment 4: Conclusions are very lengthy. They need to be crisp and point wise.
→ We appreciate your valuable indication. We have made editorial changes.
“Recently, shape memory polymers have received extensive attention because of their unique deformation mechanisms and diverse shape-changing behaviors. To meet these growing demands on renewable sources, this study aimed to synthesize a novel shape memory thermoplastic polyurethane (SMTPU) elastomer from bio-based materials. For this purpose, bio-based polyol, and 1,3-propanediol, and aromatic diisocyanate were combined via the prepolymer method with bio contents of 66−73%. Upon the investigation of various parameters on the shape memory characteristics of the obtained specimens, it was found that all bio-based SMTPUs were characterized by high weights fraction (0.77−0.92) of hydrogen-bonded urethane groups (Xb) independent of the hard segment content present in the sample. In addition, only ~0.05–0.99 wt% of the hard segment was found in the soft phase (W′2), demonstrating a high degree of phase separation in all samples, although this was particularly apparent for the MP 1/2.22/1.22 specimen. Following observation of the prepared SMTPUs by atomic force microscopy, it was found that when the content of the hard segment was increased, the hard segment domains became larger. As a result, the surface roughness and the extent of phase separation increased, leading to the hard domain structures becoming more pronounced and visible in the sample morphology. Furthermore, following evaluation by thermal and mechanical analysis, the relatively constant storage modulus indicated that no phase transitions occurred within the examined temperature range due to effective phase separation. Based on these results, a transition temperature range of 60–70 °C was selected to assess the shape memory properties of the specimens. Increase in the content of the hard segment from 30 to 40 wt% resulted in a corresponding increase in tensile strength from 23.9 to 27.3 MPa, which was accompanied by a decrease in the elongation at break from 552 to 358%, and an increase in the hardness value from 82 to 92 Shore A. Subsequently, cyclic tensile tests confirmed the reproducibility of the long-term performances of the SMTPUs, since the maintenance of elastomeric properties is of particular importance for a material even after three-dimensional (3D) printing. Moreover, shape memory analysis demonstrated that the bio-based SMTPU films and 3D-printed auxetic RE became completely fixed, and following subsequent reheating, they quickly re-covered their original shapes (recovery ratios = 100%). These results indicated that the prepared samples were not only able to fully recover their initial shapes, but that their recovery speeds were significantly shorter than those of previously reported SMTPUs. In the future, it is expected that this strategy will be extended to advanced 3D/4D printing technologies, in addition to being employed in the preparation of rapid and sophisticated reversible shape memory devices.”
Comment 5: Furthermore, following evaluation by differential scanning calorimetry and dynamic mechanical analysis, the relatively constant storage modulus indicated that no phase transitions occurred within the examined temperature range due to effective phase separation. Explain how the authors arrived at this conclusion?
→ We appreciate your valuable indication. Relevant information is described in section 3.3.2.
The well phase separation means that the division of the soft segment and hard segment regions is clear, so that the transition region of each region is clearly displayed, and the storage modulus value is maintained constant in each transition region. The storage modulus value was kept constant in the DMA diagram in the range of 45 to 70 °C corresponding to the melting temperature of soft segment on DSC. In this way, if the storage modulus appears constant in the region where melting of soft segment occurs, phase transition between hard segment and soft segment does not occur in this temperature range, and only melting of soft segment occurs while hard segment is maintained, and shape change easily occurs.
Comment 6: Moreover, shape memory analysis demonstrated that the bio-based SMTPU films became completely fixed in the form of a helical shape, and following subsequent reheating, they quickly recovered their original straight shapes. What is the reason for this behaviour?
→ We appreciate the comment.
First, in order to simply test the shape recovery performance, a film was manufactured using a press in the laboratory. Prior to the test, I looked through many references and found the simplest form to represent in two-dimensional (2D). For a vivid description of the shape memory properties, samples are used to conduct a macroscopical experiment. Even if a more complicated shape was not confirmed, it was necessary to check the shape memory performance once again by printing with a auxetic re-entrant structure in 3D printing, so it was necessary to check it in a simple film form.
POLYMER SCIENCE, SERIES B 62(5) 202 |
Macromolecules, 51(3), 705-715. |
Reactive and Functional Polymers 179,2022, 105374 |
Adv. Eng. Mater. 2022, 2201309 |
ACS Appl. Mater. Interfaces 2020, 12, 17979−17987 |
Comment 7: Figure 8. please make the legends and markings readable. It is too small.
→ Thank you for your valuable comment. We have made editorial changes as suggested.
Comment 8 In addition, the relative crosslinking densities of the polyurethane species were as-sociated the observed storage moduli in the rubbery plateau region, wherein a higher crosslinking density was associated with a greater hard segment content, indicating that the chain segments were more closely restricted by net points to inhibit their mobility How is the above statement confirmed experimentally?
→ Thank you for your valuable comment. Relevant information is described in section 3.1, 3.3.1, and 3.3.2.
Cross-linking in the hard segment of TPU is formed by hydrogen bonding between molecular chains, as mentioned in the previous article [Jung, Yang-Sook, et al (2022) One-Shot Synthesis of Thermoplastic Polyurethane Based on Bio-Polyol (Polytrimethylene Ether Glycol) and Characterization of Micro-Phase Separation. Polymers, 14(20), 4269.]. The part where these hydrogen bonds are increased forms hard segment, and the phase separation between the hard segment and the soft segment is well. When temperature is applied to TPU with well phase separation, the storage modulus has a constant value in the rubbery plateau region where the soft segment moves. References supporting this theory are listed below, and in this paper, the degree of hydrogen bonding was calculated as Xb value in FTIR, and the hard segment fraction (f) and the remaining hard segment fraction (W'2) in the soft segment was calculated and compared with the DSC and DMA result values. As a result, it was confirmed in AFM that the more hydrogen bonds were formed, the clearer the phase separation was, and in the DSC picture, it was confirmed that a rubbery plateau was formed in which the storage modulus was kept constant as a result of DMA analysis in the temperature range corresponding to the melting of soft segment.
Reference
Lei W, Fang C, Zhou X. Cheng Y, Yang R, Liu D (2017) Morphology and thermal properties of polyurethane elastomer based on representative structural chain extenders. Thermochim. Acta 653: 116–125
Van der Schuur M, Gaymans J (2006) Segmented block copolymers based on poly(propylene oxide) and mono disperse polyamide-6, T segments. J. Polym. Sci. Part A Polym. Chem., 44:4769–81
Peebles H (1974) Sequence Length Distribution in Segmented Block Copolymers. Macromolecules 7: 872–882
Flory J (1942) Thermodynamics of High Polymer Solutions. J. Chem. Phys 10: 51

Round 2
Reviewer 2 Report
Accept